# The Performance of Different Machine Learning Algorithm and Regression Models in Predicting High-Grade Intracranial Meningioma

**DOI:** 10.3390/brainsci13040594

**Published:** 2023-03-31

**Authors:** Haibo Teng, Xiang Yang, Zhiyong Liu, Hao Liu, Ouying Yan, Danyang Jie, Xueying Li, Jianguo Xu

**Affiliations:** 1Department of Neurosurgery, West China Hospital, Sichuan University, Chengdu 610044, China; 2Department of Oncology, Cancer Center, West China Hospital, Sichuan University, Chengdu 610041, China

**Keywords:** high-grade meningioma, nomogram, logistic regression, machine learning

## Abstract

Meningioma is the most common primary tumor of the central nervous system (CNS). Individualized treatment strategies should be formulated for the patients according to the WHO (World Health Organization) grade. Our aim was to investigate the effectiveness of various machine learning and traditional statistical models in predicting the WHO grade of preoperative patients with meningioma. Patients diagnosed with meningioma after surgery in West China Hospital and Shangjin Hospital of Sichuan University from 2009 to 2016 were included in the study cohort. As the training cohort (n = 1975), independent risk factors associated with high-grade meningioma were used to establish the Nomogram model. which was validated in a subsequent cohort (n = 1048) from 2017 to 2019 in our hospital. Logistic regression (LR), XGboost, Adaboost, Support Vector Machine (SVM), K-Nearest Neighbor (KNN), and Random Forest (RF) models were determined using F1 score, recall, accuracy, the area under the curve (ROC), calibration plot and decision curve analysis (DCA) were used to evaluate the different models. Logistic regression showed better predictive performance and interpretability than machine learning. Gender, recurrence history, T1 signal intensity, enhanced signal degree, peritumoral edema, tumor diameter, cystic, location, and NLR index were identified as independent risk factors and added to the nomogram. The AUC (Area Under Curve) value of RF was 0.812 in the training set, 0.807 in the internal validation set, and 0.842 in the external validation set. The calibration curve and DCA (Decision Curve Analysis) indicated that it had better prediction efficiency of LR than others. The Nomogram preoperative prediction model of meningioma of WHO II and III grades showed effective prediction ability. While machine learning exhibits strong fitting ability, it performs poorly in the validation set.

## 1. Introduction

Meningioma has an annual incidence of approximately 1/100,000 and is currently considered the most common type of intracranial primary central nervous system (CNS) neoplasms, accounting for 25–30% [1]. According to the WHO classification in 2016, it is divided into three levels, of which WHO I is less invasive and WHO II and WHO III are more invasive [2]. The meningiomas of WHO II and III grade account for about 18% of cases, which is more aggressive and leads to a worse clinical prognosis [3,4]. Although meningioma is considered a benign tumor, the recurrence and invasion of high-grade meningioma cannot be ignored.

Numerous variables have been utilized to predict the prognosis of meningioma. To overcome the limitation of WHO grade, some researchers took advantage of DNA methylation to propose a new classification of meningioma [5]. The WHO grade is still the mainstream at present, so it is crucial for surgeons to predict pathological grades through a non-invasive scheme. Previous models can rarely be applied, and the selection of variables is not uniform because of, for example, insufficient sample size or rigorous screening method [6], or overfitting results. Some models require particular indicators, such as multimodal MRI, gene detection, and other data that are difficult to obtain [7,8]. With the development of artificial intelligence, machine learning plays a large part in the research of meningioma prediction. Especially with the birth of imageomics, researchers have established different models through MRI or other specific images to predict clinical results. The use of surgical specimens for detection after surgery limits the early diagnosis and intervention of WHO II and III meningiomas. Therefore, we need a brief model that can accurately judge the invasion of tumors before the operation to guide the decision of the surgeon during an operation

To overcome the above limitations, we combined data from two hospitals and proposed a new model to predict the case grade of meningioma, and verified it at different time points and institutions. To attain the most effective model, we used six different machine learning methods, such as logistic regression (stepwise back-off method), XGboost (eXtreme Gradient Boosting), Adaboost (Adaptive boosting), SVM (Support Vector Machine), KNN (K-NearestNeighbor) and RF (Random forest) to screen variables and build models. The purpose of this study is to provide a reliable basis for the surgical management and scheme of high-grade meningioma.

## 2. Method

### 2.1. Patients

The workflow is summarized in Figure 1. After the approval of the Ethics Committee of West China Hospital, data were collected by Haibo Teng, Danyang Jie, Ouying Yan, Xueying Li, and some medical students through the HIS system (Hospital Information System), PACS system (Picture Archiving and Communication Systems), LIS system (Laboratory Information System) and pathology system. A total of 5385 patients with postoperative pathological diagnosis of meningioma from 2009 to 2019 were included in this study. Data were screened by Xiang Yang and Zhiyong Liu, and the exclusion criteria were as follows: (1) Pathological results failed to identify meningioma, or there were disputes (n = 120); (2) Pathological results were unable to identify WHO grade of meningioma (n = 22); (3) General clinical Data (including test results and imaging results) were incomplete or missing (n = 2086). Finally, 3023 complete cases were included: 1975 patients in the training group from 2009 to 2016 and 1048 patients in the validation group from 2017 to 2019.

### 2.2. Variable Definition

We included basic medical history data (age, gender, recurrence, history, blood type), imaging data (tumor size, shape, location, boundary, skull invasion, dural tail sign cystic, edema, MRI signal intensity CT), laboratory indicators (hemoglobin, platelet, white blood cell, neutrophils, lymphocyte, monocyte, eosinocyte, basicytebasophil, dural tail sign, albumin to globulin ratio, lymphocyte to monocyte ratio, platelet to lymphocyte ratio, neutrophil to lymphocyte ratio). Tumor size is defined as the largest diameter of the largest section of the tumor on MRI. Clear tumor boundary refers to the presence of punctate or cord-like vascular shadows and annular cerebrospinal fluid signal bands between the tumor and brain tissue. Peritumoral edema was defined as the largest diameter of the area measured at the maximal section of the edema at the tangent to the tumor edge on T2 and Flair-weighted images. Skull involvement includes skull compression, bone resorption, skull thinning, skull reactive hyperplasia, and skull invasion by the tumor on CT. Cyst is defined as a non-enhancing hypodense circular or nearly circular area with signal intensity lower than the tumor region on T1 weight images and higher than the tumor region on T2 weight images. According to the previous literature, we divided tumor locations into four categories: supratentorial, cerebellum and tentorium, skull base, and ventricle. Through the evaluation and discussion of two radiologists, the normal MRI signal intensity was divided into four categories: low, medium, high, and mixed, and the enhanced MRI was divided into two categories: uniform and inhomogeneous signals.

### 2.3. Algorithm Training and Evaluation

To impute missing data, multiple imputations were performed using multivariate imputations by missForest, and variable screening was performed using R (3.5.3 version) for logistic regression and machine learning analysis. Data were pre-processed, and the Chi-square test or Mann—Whitney-U was performed to confirm the relative impact of single factors. Collinear (Variance Inflation Factor, i.e., VIF > 10) and highly correlated variables (Kendall *p* > 0.5) were excluded by computing the correlation coefficients. LASSO regression, logistic regression (stepwise back-off method), XGboost (eXtreme Gradient Boosting), Adaboost (Adaptive boosting), SVM (Support Vector Machine), KNN (K-NearestNeighbor) and RF (Random forest) was used to identify factors associated with meningiomas of WHO II and III grade. K-fold cross-validation for model selection and tuning is based on k-fold cross-validation with k = 10. Model optimization and selection process only used a training set. Confusion matrix metrics of precision (positive predictive value), recall (sensitivity), F1 (harmonic mean between precision and recall), DCA, and AUC were computed to compare models. Nomogram was established after the variable screening and model selection. The model efficacy in the external cohort was judged by using the AUC. *p* values of <0.05 were considered statistically significant.

## 3. Result

Compared with males, females were more likely to develop meningioma (F:M = 2.48:1), but males had a higher incidence rate of meningiomas of WHO II and WHO III (51% vs. 49%). The baseline characteristics of the patients and the result of the single analysis are shown in Table 1. The VIF of all factors was less than 10, and the heat map (Figure 1A) showed the correlation between variables. The result of LASSO regression is shown in Figure 1B,C. Common factors were selected after building five machine-learning models and performing traditional regression. The order of factor importance in each model is shown in Figure 1D–H. We ranked the top ten features of feature importance in all machine learning methods and compared them with the linear model of logistic regression. Finally, based on the single-factor analysis, the factors with high importance in the six model methods (including LOSSA) were determined to be included in the final model comparison. Gender, recurrence history, T1 signal intensity, enhanced signal degree, peritumoral edema, tumor diameter, cystic, and the location was selected into the final model. Table 2 lists the evaluation metrics of all the prediction models, and Figure 2A,B displays the comparison of ROC curves for the six models. In the training set, XGboost and RandomForest showed the highest AUC value (=0.999). The AUC values of SVM, KNN, and Adaboost are 0.675, 0.886, and 0.855, respectively (Figure 3). The AUC values of logical regression are 0.812. In the validation set, the AUC values of XGboost and RandomForest are only 0.741 and 0.748. The AUC values of SVM, KNN, and Adaboost are 0.652, 0.666, and 0.779, respectively. The ACU value of logical regression is 0.807. Figure 2C,D displays calibration and DCA plots of six models in an internal validation set. A nomogram according to the result of variable selection was established by LR (Figure 4A). Figure 4B,C displays the AUC (0.842) and calibration plot in external validation. The model established by LR demonstrated good performance in both external and internal validation.

## 4. Discussion

Most meningiomas are considered to have benign biological behavior, but high-grade meningiomas were not uncommon. Surgery is the best treatment for meningioma at present, and the choice of surgical methods [9] and tumor grade are related to the clinical prognosis. The largest prospective longitudinal series of long-term HRQoL results after surgery reported sustained and clinically significant impairments [10]. Although gross total resection (GTR) is the goal that neurosurgery pursues, the balance of tumor progression and quality of life is extremely important. It is incorrect to use the same treatment strategy and surgical approach for patients with meningioma of different pathological grades, especially at critical neurovascular sites. Neurosurgeons are primarily judged empirically with imaging, which is not comprehensive. In our study, we found that the patient’s basic information and blood indicators may be biomarkers of the pathological grade of potential meningiomas. We consequently set up a more comprehensive model to guide treatment options for the operating surgeon.

In our study, gender, recurrence history, T1 signal intensity, enhanced signal degree, peritumoral edema, tumor diameter, cystic, and location are used to build the final model by logistic regression. Although the machine learning methods such as XGBoost and RandomForest have shown absolute advantages in the training set (AUC = 0.999), it has shown very low effectiveness in the validation set (AUC = 0.741 and 0.748, relatively). To the best of our knowledge, we are the first prediction model of meningioma grade established by variable methods of factors screening and statistical analysis with large sample queues from the different medical centers. Shijun Peng et al. constructed a prediction model with an internal concordance index of 0.868 [6], but only 200 sample sizes and uncritical variable selection cannot establish an accurate model. Radiomics is a trend to develop models at present. However, in the current situation, the diversity of imaging machines cannot reach the same consistency as the objective blood indicators, which leads to the limitations of the research on imageomics in clinical applications. Although Roshan Karri et al. performed popular machine learning to construct the model with an AUC of 0.8–0.9 [11], the lack of external verification and inexplicability makes machine learning not necessarily the best choice for every model. The advantage of machine learning is that it integrates the resources of the training set to the maximum extent through intelligent algorithms, thus ensuring that the resulting model is highly effective. However, the biggest problem is that machine learning may produce over-fitting. To prevent this, an effective and rigorous machine learning model should be built with multi-center, multi-dimensional, and large samples. However, the present research has been unable to accomplish this. Therefore, in this case, we should not abandon the advantages of traditional statistics.

The lack of a rigorous modeling process has been pointed out as the disadvantage of most neurosurgery model studies [12]. To evaluate whether the population has enough credibility to establish the model, we perform the sample size calculation. We conducted four assessments about sample size [13] and utilized an interactive tool (https://mvansmeden.shinyapps.io/BeyondEPV/ (accessed on 1 October 2021)). Based on the 30 candidate variables included in the study, the minimum sample size required for new model development was 1640 patients, with 246 events at least and a minimum of 8.2 events per predictor parameter. Thus, the actual sample of 1975 patients in the development cohort and 310 events of WHO II and III grades likely provided sufficient power to ensure the credibility of our models.

In the current study, the patients for validation came from a patient’s cohort at different times and geographic locations. We selected the possible independent risk factors of WHO II and III grades by comparing the five machine learning algorithms and the LR. The AUC of all classifiers had a statistically significant difference (*p* = 0.004). Although the machine learning model achieved comparable precision to LR in internal validation, the performance of the LR model was more prominent in external validation, including the recall, F1, and AUC. This may be caused by the overfitting of machine learning. Therefore, we chose LR to establish the final prediction model. To make the model of logical regression more intuitive and clinical application, we showed it through the nomogram.

Compared with other models, we have included many variables and comprehensively considered various algorithms in the screening process to obtain the best correlation. On the other hand, we used data from multiple centers and large samples for research and conducted external validation, which effectively made the model more universal and put an end to over-fitting. The simple model established in this study may guide the surgical strategies of neurosurgeons. Tumor size, edema, and recurrence history were the most important risk factors for WHO II and III grades and caused poorer clinical prognosis [14,15,16]. Previous studies also used the tumor volume and peritumoral edema volume to predict the degree of invasion [17]. For patients with large diameter and severe edema, even in the important and complex neurovascular areas such as the sellar region and sphenoid ridge, gross total resection should be conducted to the greatest extent to reduce the probability of recurrence. Male patients with meningioma in convex locations were more likely to have meningiomas of WHO II and III grade, which required that neurosurgeons should try their best to perform gross total resection. Boukobza M et al. found that the proportion of WHO II and III grades in cystic meningiomas was 23% [18]. We also found that the cystic change increased the possibility of a higher grade, but there were still few studies on meningioma with cysts. When the model is used to evaluate preoperative patients, if the recurrence probability of patients is less than 0.5, neurosurgeons should pay more attention to the impact of surgery on the quality of life of patients rather than pursue the perfection of surgery. If the recurrence probability of the patient is high, then the surgeon should try his best to perform total resection of the tumor to prolong the progression-free survival time of the patient.

Several limitations should be acknowledged. Our study is susceptible to selection and reporting bias inherent as for the other retrospective research. Secondly, although our validation comes from different hospitals and periods, which increases the generalization of the model to a certain extent, it also leads to potential selection bias. Therefore, in the future, more and more patients in the center need to be included in the model validation. Third, we lack long-term and continuous HRQoL questionnaire follow-up for patients, which makes the relationship between surgical management and postoperative quality of life unclear. Fourth, we have not included other new technologies, such as multimodal MRI, in the model due to the large span of years. Even so, the model also yielded satisfactory discrimination and calibration in a cohort.

## 5. Conclusions

In this study, a multi-center, large-sample meningioma database was established, and a prediction model of meningioma grade was established using logical regression and five different machine-learning methods. Finally, the site was selected as the predictor of meningioma grade, and it was found that the traditional logistic regression model performed better in the training set and validation set. We believe that the model can guide clinical decision-making and improve the prognosis of patients. At the same time, the model also needs a larger external verification queue for research.

## Figures and Tables

**Figure 1 brainsci-13-00594-f001:**
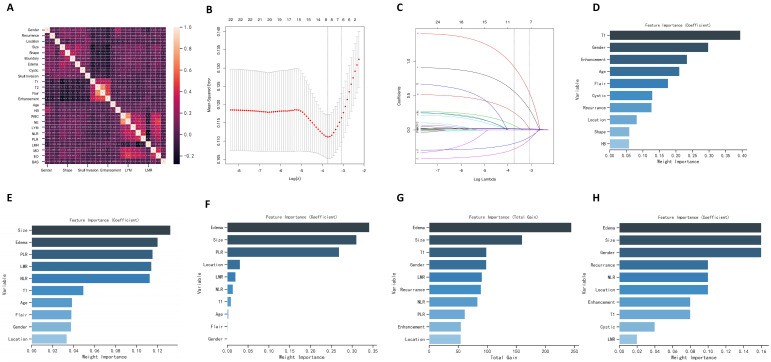
The selection of risk factors related to high−grade intracranial meningioma. (**A**) Heat map showing the correlation with risk factors. (**B**,**C**) Feature variable selection using least absolute shrinkage and selection operator (LASSO) regression in the development cohort. (**D**−**H**) Feature importance in Support Vector Machine (**D**), Random Forest (**E**), K-Nearest Neighbor (**F**), XGboost (**G**), and Adaboost (**H**) model.

**Figure 2 brainsci-13-00594-f002:**
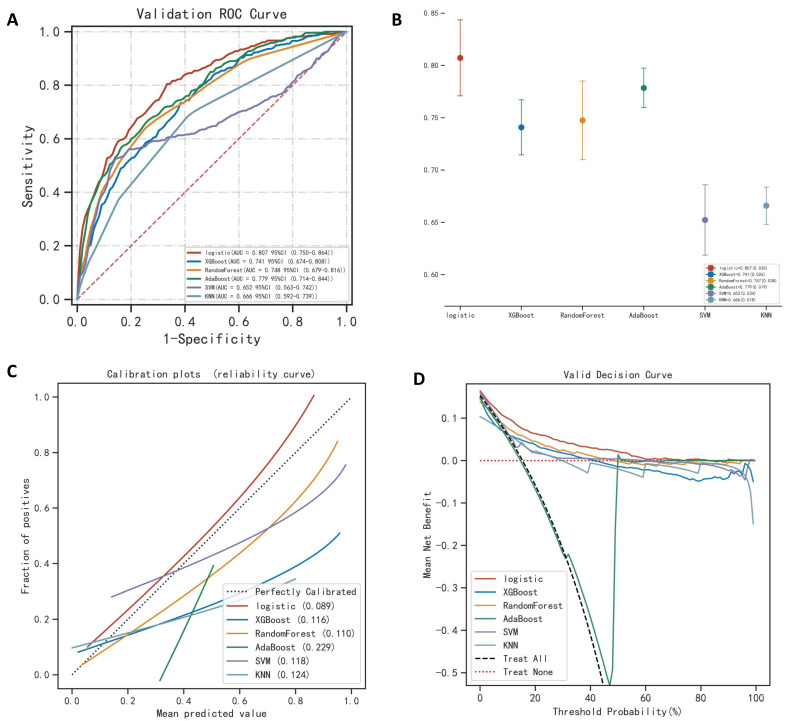
The performance of Logistic regression (LR), XGboost, Adaboost, Support Vector Machine (SVM), K- Nearest Neighbor (KNN), and Random Forest (RF) model in internal validation. (**A**,**B**) Receiving operating characteristic curves and forest plots showing the performance of six models in the internal validation cohort. (**C**) Calibration curves for testing the stability of six models in the external validation cohorts. (**D**) Decision curve analysis of six models for the internal validation cohorts.

**Figure 3 brainsci-13-00594-f003:**
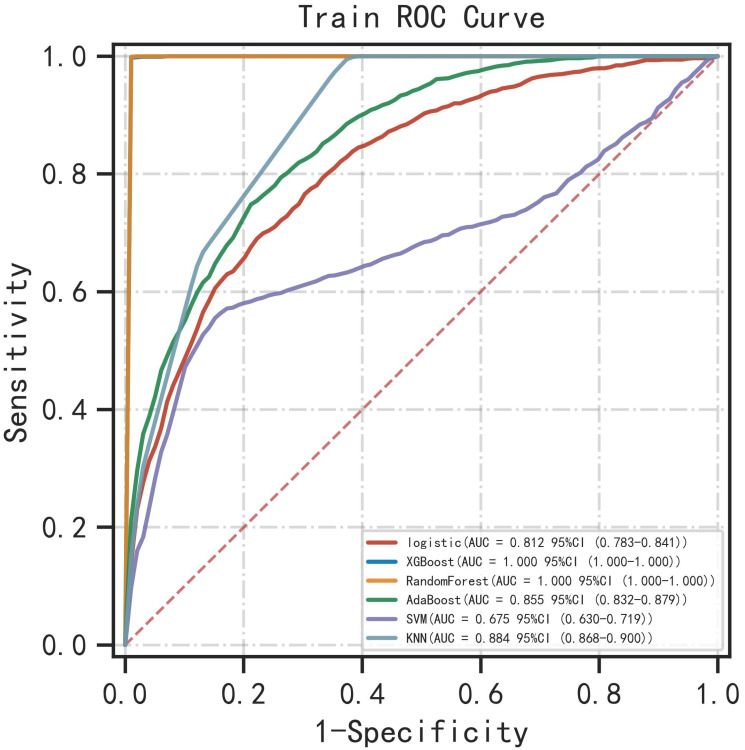
The performance of Logistic regression (LR), XGboost, Adaboost, Support Vector Machine (SVM), K-Nearest Neighbor (KNN), and Random Forest (RF) model in the training cohort.

**Figure 4 brainsci-13-00594-f004:**
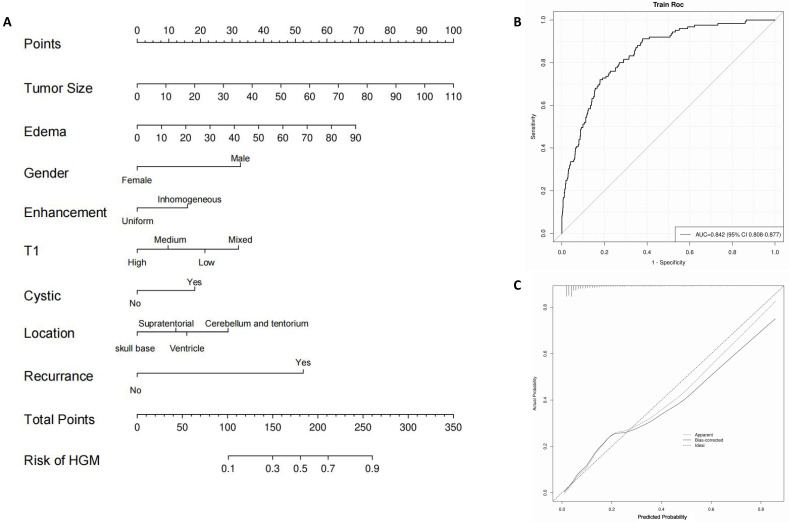
Development and application of a model for predicting high-grade intracranial meningioma (HGM). (**A**) The prediction model. (**B**) Receiving operating characteristic curves showing the performance of the prediction model in the external validation cohort. (**C**) Calibration curves for testing the stability of the prediction model in the external validation cohorts.

**Table 1 brainsci-13-00594-t001:** Baseline characteristics of patients with meningioma.

Variable	Meningioma			*p* Value
	Overall	WHO I	WHO II and III	
Age(year), n(%)				
0–20	25 (1.266)	14 (0.841)	11 (3.548)	<0.001
20–40	272 (13.772)	220 (13.213)	52 (16.774)	
40–60	1074 (54.380)	920 (55.255)	154 (49.677)	
>60	604 (30.582)	511 (30.691)	93 (30.000)	
Gender, n(%)				
Female	1407 (71.241)	1256 (75.435)	151 (48.710)	<0.001
Male	568 (28.759)	409 (24.565)	159 (51.290)	
Recurrance, n(%)				
No	1905 (96.456)	1635 (98.198)	270 (87.097)	
Yes	70 (3.544)	30 (1.802)	40 (12.903)	
Boundary, n(%)				
Clear	1751 (88.658)	1493 (89.670)	258 (83.226)	0.001
Unclear	224 (11.342)	172 (10.330)	52 (16.774)	
Shape, n (%)				
Regular	1243 (62.937)	1107 (66.486)	136 (43.871)	<0.001
Irregular	732 (37.063)	558 (33.514)	174 (56.129)	
Location, n (%)				
Supratentorial	383 (19.392)	348 (20.901)	35 (11.290)	
Cerebellum and tentorium	1217 (61.620)	981 (58.919)	236 (76.129)	
Skull base	295 (14.937)	269 (16.156)	26 (8.387)	
Ventricle	80 (4.051)	67 (4.024)	13 (4.194)	
T1, n (%)				
Low	361 (18.278)	239 (14.354)	122 (39.355)	
Medium	533 (26.987)	454 (27.267)	79 (25.484)	
High	1059 (53.620)	952 (57.177)	107 (34.516)	
Mixed	22 (1.114)	20 (1.201)	2 (0.645)	
T2, n (%)				
Low	593 (30.025)	436 (26.186)	157 (50.645)	
Medium	145 (7.342)	129 (7.748)	16 (5.161)	
High	605 (30.633)	540 (32.432)	65 (20.968)	
Mixed	632 (32.000)	560 (33.634)	72 (23.226)	
Flair, n (%)				
Low	599 (30.329)	446 (26.787)	153 (49.355)	
Medium	191 (9.671)	160 (9.610)	31 (10.000)	
High	534 (27.038)	479 (28.769)	55 (17.742)	
Mixed	651 (32.962)	580 (34.835)	71 (22.903)	
Enhancement, n (%)				
Yes	1231 (62.329)	1115 (66.967)	116 (37.419)	
No	744 (37.671)	550 (33.033)	194 (62.581)	
HBG, n (%)				
Normal	1532 (77.570)	1313 (78.859)	219 (70.645)	0.001
Abnormal	443 (22.430)	352 (21.141)	91 (29.355)	
PLT, n (%)				
Normal	1700 (86.076)	1437 (86.306)	263 (84.839)	0.493
Abnormal	275 (13.924)	228 (13.694)	47 (15.161)	
WBC, n (%)				
Normal	1747 (88.456)	1490 (89.489)	257 (82.903)	<0.001
Abnormal	228 (11.544)	175 (10.511)	53 (17.097)	
NEUT, n (%)				
Normal	1705 (86.329)	1460 (87.688)	245 (79.032)	
Abnormal	270 (13.671)	205 (12.312)	65 (20.968)	
LYM, n (%)				
Normal	1667 (84.405)	1425 (85.586)	242 (78.065)	
Abnormal	308 (15.595)	240 (14.414)	68 (21.935)	
MO, n (%)				
Normal	1750 (88.608)	1492 (89.610)	258 (83.226)	0.001
Abnormal	225 (11.392)	173 (10.390)	52 (16.774)	
EO, n (%)				
Normal	1679 (85.013)	1436 (86.246)	243 (78.387)	<0.001
Abnormal	296 (14.987)	229 (13.754)	67 (21.613)	
BAS, n (%)				
Normal	1918 (97.114)	1624 (97.538)	294 (94.839)	0.009
Abnormal	57 (2.886)	41 (2.462)	16 (5.161)	
Blood Type				
O	666 (33.722)	564 (33.874)	102 (32.903)	0.47
A	641 (32.456)	529 (31.772)	112 (36.129)	
B	495 (25.063)	423 (25.405)	72 (23.226)	
AB	173 (8.759)	149 (8.949)	24 (7.742)	
Sull Invasion, n (%)				
NO	1654 (83.747)	1415 (84.985)	239 (77.097)	<0.001
YES	321 (16.253)	250 (15.015)	71 (22.903)	
Cystic, n (%)				
NO	1785 (90.380)	1544 (92.733)	241 (77.742)	
YES	190 (9.620)	121 (7.267)	69 (22.258)	
DTs ,n (%)				
NO	426 (21.570)	351 (21.081)	75 (24.194)	0.221
YES	1549 (78.430)	1314 (78.919)	235 (75.806)	
AGR, median [IQR]	1.594 [1.439,1.776]	1.598 [1.446,1.774]	1.557 [1.410,1.779]	0.063
LMR, median [IQR]	5.063 [3.833,6.697]	5.172 [3.918,6.783]	4.649 [3.286,6.077]	<0.001
PLR, median [IQR]	105.882 [79.755,139.091]	103.349 [79.359,136.567]	115.169 [82.386,156.571]	
NLR, median [IQR]	2.117 [1.620,3.010]	2.067 [1.571,2.891]	2.484 [1.886,4.236]	
edema, mean (±SD)	11.718 ± 15.877	9.674 ± 14.411	21.113 ± 18.674	
size, mean (±SD)	43.943 ± 18.070	41.289 ± 16.792	56.003 ± 18.747	

HBG, hemoglobin; PLT, platelet; WBC, white blood cell; NEUT, neutrophils;LYM, lymphocyte; MO, monocyte; EO, eosinocyte;BAS, basicytebasophil; DTs, dural tail sign; AGR, albumin to globulin ratio; LMR, lymphocyte to monocyte ratio; PLR, platelet to lymphocyte ratio; NLR, neutrophil to lymphocyte ratio.

**Table 2 brainsci-13-00594-t002:** Metrics of different prediction models.

Dataset	Model Performance	Method					
		Logistic	XGBoost	RandomForest	AdaBoost	SVM	KNN
Train	AUC (SD)	0.812 (0.009)	1.000 (0.000)	1.000 (0.000)	0.855 (0.007)	0.675 (0.007)	0.884 (0.006)
	Accuracy (SD)	0.777 (0.016)	0.996 (0.001)	0.997 (0.001)	0.775 (0.020)	0.803 (0.010)	0.841 (0.006)
	Recall (SD)	0.679 (0.043)	0.998 (0.003)	0.997 (0.003)	0.774 (0.026)	0.559 (0.031)	1.000 (0.000)
	F1 (SD)	0.485 (0.008)	0.991 (0.005)	0.997 (0.001)	0.524 (0.012)	0.470 (0.018)	0.667 (0.009)
Validation	AUC (SD)	0.807 (0.036)	0.741 (0.026)	0.748 (0.038)	0.779 (0.019)	0.652 (0.034)	0.666 (0.018)
	Precision (SD)	0.774 (0.036)	0.815 (0.009)	0.842 (0.014)	0.740 (0.010)	0.773 (0.014)	0.775 (0.011)
	Recall (SD)	0.804 (0.077)	0.690 (0.095)	0.642 (0.084)	0.694 (0.078)	0.542 (0.076)	0.646 (0.128)
	F1 (SD)	0.525 (0.061)	0.493 (0.049)	0.549 (0.051)	0.430 (0.019)	0.430 (0.034)	0.402 (0.046)

SVM, support Vector Machine; KNN, k-nearest-neighbors; SD, standard deviation; AUC, area under the receiver operating characteristic curve.

## Data Availability

The datasets used for the current study are available from the corresponding author upon reasonable request.

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
