# Peer review of "The Performance of Different Machine Learning Algorithm and Regression Models in Predicting High-Grade Intracranial Meningioma"

_brainsci, 2023, doi:10.3390/brainsci13040594_

Round 1
Reviewer 1 Report
Nice, short and well structured analysis on the effectiveness of various machine learning versus traditional statistical models employed to predict grades of preoperative patients with meningioma.
The manuscript contains original results and it is written in a very clear way. The research design is appropriate and the methodology employed is adequately described. It meets the criteria of scientific quality and relevance for this journal.
I propose that the Authors take care of a minor aspect connected to style and presentation. The acronym WHO, even if well known, should be made explicit in the abstract. The same for acronyms from line 101 to 109.
Author Response
1.I propose that the Authors take care of a minor aspect connected to style and presentation. The acronym WHO, even if well known, should be made explicit in the abstract. The same for acronyms from line 101 to 109.
Re: Thank you for pointing out that we have explained the relevant abbreviations according to your suggestions
Reviewer 2 Report
This is an interesting paper that adds to the current knowledge base and is appropriate for the Brain Sciences Journal. It is my understanding that the aim was to evaluate different ML models for prediction of high-grade meningioma preoperatively.
Recommended revision to improve this manuscript:
1. There are numerous grammar and spelling mistakes.
2. Introduction, line 42-43: “Some molecular characteristics… are believed that it can be used to create a new meningioma grade” - can be worded better.
3. In the methods section, list all data collected and do not use “etc”. For example, list all medical data, all imaging data, and all lab data.
- In the methods section, there is some vagueness about who is selecting your data. Line 59 - “data was collected by professionals.” Who are the professionals e.g., medical students? Line 63 - “data were screened by 2 experts.” Who are the experts?
- Patients from the training group and the validation group are from two different time points. I know it was discussed later in the paper that using different time points and geography improves generalizability, but it can also be an opportunity for confounding. That can be discussed.
- Clarify exactly which images are used to define edema, cysts, tumor, bone changes.
- In algorithm training and validation, line 100 - VIF is undefined anywhere in the paper.
- Line 165 - did you mean “screening factors” and not “factors screening”?
- Claim that predicting grade through routine sequences and blood indicators more suitable than radiomics (line 168-170). What is the source of that information?
- The literature typically defines meningioma by WHO grade 1, 2, or 3. The use of LGM and HGM is not standard. Recommend reformatting your demographic table and divide data by WHO grade and do not use terms LGM and HGM throughout analysis and discussion.
- The discussion of the literature on machine learning for meningioma should be expanded and more comprehensive. The paper can limit discussion on other topics not germane to the primary thesis of this manuscript.
- The discussion should conclude with the primary finding. List the exact variables needed and the exact model needed for your best recommendation.
- The discussion should better define how this result improves patient care.
- Add a note on Author Contributions, Funding, Institutional Review Board, Informed Consent, Acknowledgements, and Author Conflicts of Interest.
Reviewer 3 Report
The authors studied the prediction of preoperative meningioma through a variety of machine learning and traditional statistical models. The authors analyzed the data of patients collected from 2009 to 2019 and selected some variables for model training. Furthermore, the authors propose a new model to predict the case grade of meningioma. I have the following comments for authors to consider:
1. the authors did not give a detailed description of the proposed model and the evaluation of the model, which are the most important parts of this work. The authors should present the model and evaluation of its performance in different ways, such as figures, formulars or tables.
2. The coordinates and labels in some figures are too small to read. For example, figure 1, figure 3, figure 4, etc. It is suggested that the author redesign and typeset these pictures.
3. For D to H in Figure 1, the authors should present the methods for obtaining the importance of the weights of the feature variables in the different models.
4. The paper introduced a lot about data processing; however, it did not mention the ratio of train and test set.
5. There are some typos and grammar mistakes in the paper.
Round 2
Reviewer 3 Report
(1) It is essential to provide details on how the weight importance of feature variables in various models was obtained, such as specific processes or related formulas, when describing the results of this paper's proposed model.
(2) In this article, the model focuses on selecting variables for training parameters, however, the results section only introduces the selected variables in a single sentence without analyzing the rationale behind the selection.
(3) In 2.3, The author mentions the optimization of the models. I suggest that the authors describe the necessary optimization process, such as adjustment of parameters, adjustment of model structure, or improvement of related algorithms.
Round 3
Reviewer 3 Report
1、The readability of Table 1 is poor, it is recommended to redesign the table.
2、There is a reference that only introduces the authors and title, and lacks other necessary information.